# Consideration of Lamination Structural Analysis in a Multi-Layered Composite and Failure Analysis on Wing Design Application

**DOI:** 10.3390/ma14133705

**Published:** 2021-07-02

**Authors:** Ernnie Illyani Basri, Mohamed Thariq Hameed Sultan, Adi Azriff Basri, Faizal Mustapha, Kamarul Arifin Ahmad

**Affiliations:** 1Department of Aerospace Engineering, Faculty of Engineering, Universiti Putra Malaysia (UPM), Serdang 43400, Selangor, Malaysia; ernebasri@gmail.com (E.I.B.); thariq@upm.edu.my (M.T.H.S.); adiazriff@upm.edu.my (A.A.B.); faizalms@upm.edu.my (F.M.); 2Aerospace Malaysia Innovation Centre (944751-A), A Prime Minister’s Department, MIGHT Partnership Hub, Jalan Impact, Cyberjaya 63000, Selangor, Malaysia; 3Aerospace Malaysia Research Centre (AMRC), Universiti Putra Malaysia (UPM), Serdang 43400, Selangor, Malaysia

**Keywords:** laminates composite, computational modeling, finite element analysis, failure assessment

## Abstract

A finite element (FE) model is developed to study the structural performance on a composite wing of a UAV with a tubercle design at the leading edge of the wing. The experimental study of the designation of the composite at the wing skin is carried out to prove the simulation validity through material characteristics. In this paper, the numerical modeling for simulation is highlighted to correlate the process parameter setting of simulation replicating the actual experimental tests. The percentage difference was calculated to be 11.1% by tensile and 10.47% by flexural. The numerical work applied the study of FE analysis and developed a standardized numerical approach for structural optimization, known as FE-ACP simulation. The significant findings of deformation are obtained according to Schrenk’s aerodynamic loading, while the prediction of failure mode of Tsai–Wu under interaction among stresses and strains was acquired at the seventh and eighth layer of both spars.

## 1. Introduction

The state-of-the-art advances in today’s composite materials and adaptive structures have made possible the idea of Unmanned Aerial Vehicle (UAV) wing to be designed and manufactured within a feasible period and at a reasonable price. The main goal is to create a modern UAV with a high strength-to-weight ratio to give an optimal performance at all stages of multidisciplinary optimization including shape parametrization, parametric formulation and constraint approximation with the aid of computational systems. In particular, finite element (FE) modeling has been widely used in the applications of civil, aerospace and mechanical engineering to determine the accurate response of the product design. It is commonly acknowledged as a computer-aided mathematical technique to determine numerical solutions based on abstract equations of calculus that predict the behavior of physical systems under external effects [1]. The FE modeling has proven to be a powerful tool used in aerospace engineering applications due to its advantage of being applicable to irregular geometry in the form of shells or plates that contain composite material properties.

In the previous studies, the application of materials on the UAV wing of high aspect ratio highlighted the structural performance, particularly, from the computational and experimental perspectives. The use of FE analysis with the aid of simulation software is applied specifically on the parts of the UAV wing with the material used, thus the optimized parameters were also highlighted.

Rajagopal and Ganguli [2] investigated the UAV wing design optimization that achieved long endurance with minimum weight using NASTRAN. The authors used a Kriging method-based FE method for static analysis on the wing of a carbon fiber-reinforced plastic composite. The pressure distribution was subjected to the aerodynamic lift and drag of optimum NASA/LANGLEY LS (1) (GA (W)-1) airfoil. Hence, the analysis suggested that the maximum value of stress was found at the wing root and maximum deflection was at the wingtip. Shabeer and Murtaza [3] studied the ideal design structure of a UAV wing and included all internal components and wing skin. The static FE analysis was carried out with the aid of NASTRAN software. In this case, the other internal components were made of alloys, whereas the wing skin was made of composite materials. The authors also conducted a similar analysis with different orientations of composite skin layers. The orientation of [0/90/+45/−45/90/0] showed an improved performance in terms of maximum von Mises stress at the root of the wing and displacement, with the values of 50.7 MPa and 4.63mm, respectively. Kanesan et al. [4] studied the deflection on the UAV wing of NACA4415 airfoil with the aid of ABAQUS software (version 6.11, Dassault Systemes Simulia Corp., Providence, RI, USA). The wing skin and internal members are made of composite materials. Higher deflection is observed near the tip of the wing, which resulted within the range of 0.35% to 16.4% compared to the bending experiment. Prabhu et al. [5] compared both taper (NACA0012) and rectangular (NACA2412) composite wings to determine the optimum structural UAV wing with a high stiffness/weight ratio and able to withstand unexpected loading. By using ANSYS software (version 12, ANSYS Inc., Southpointe, Canonsburg, PA, USA), the static FE analysis is conducted on both wings. In the study, the analyses were performed by varying the different number of internal components of the wing (stringers and ribs) and materials composition (alloy and composites). Hence, the study highlighted that stringers and ribs contributed to the stress acting on the entire wing structure. Kavya and Reddy [6] carried out the static, modal and buckling analyses of FE modeling using ANSYS software. All analyses were performed with the air pressure applied on three different composite materials including with/without spars and ribs. From the study, it can be concluded that adding spars and ribs contributed to the high strength and less weight of the S Glass material. Ramos [7] performed a static FE analysis on a long endurance electric UAV wing structure by using CATIA Generative Structural software (version V5, Dassault Systemes Simulia Corp., Providence, RI, USA). In the study, the wing skin was made of fiberglass composite, whereas the spar and ribs were made of balsa wood. The maximum deformation and von Mises stress are found at the end of the spar. The results satisfied the requirement of material strength of each structure.

On the other hand, the experimental structural analysis involved the testing hypotheses to be matched on the actual structural behavior of the wing structure. Rowe [8] experimented on static structural analysis of a simplified inflatable wing of NACA4318 airfoil. The wing was made of urethane-coated Vectran material. The experimental static bending test was set up in such a way that the wing was mounted to a rigid test stand with vertical loads from the wingtip and mid-span of trailing edges. The test proved that wing deflection decreased with increasing internal pressure at the highest internal pressure, hence deflection was observed at the wingtip. Sullivan et al. [9] measured the vibration and static response characteristics of carbon composite materials of ultralight UAV. A whiffletree system was used for static loading subjected to the distribution by a high-g pull-up maneuver. The maximum deformation was at the fore spar, and wing skin was buckled at various locations before reaching the limit load. A similar experiment was also performed by Sullivan et al. [10] but focused on the variation of the laminate orientation of each wing component. In the study, the wing skin was made of a foam core, while other internal components were made of carbon fiber/epoxy. The composite wing was attached to the saddle fixture in an upside-down position and the loads were applied downward using a three-tier whiffletree system. The wing showed the initial failure at the left-wing, which exceeded its design ultimate load, particularly at the aft spars. Paradies and Ciresa [11] conducted static and dynamic loads on a thin composite curved wing of glass fiber and foam core, which was attached with roll control of the piezoelectric element. No spar or ribs were acquired in these experiments. The active wing showed a deflection on the trailing edge near the wing root. Hence, the author concluded that the deformation obtained was sufficient enough to balance with roll control, with results in good agreement obtained between computational and experimental results. Gaunt et al. [12] conducted two tests on the UAV NACA 2415 airfoil, with wing skin of carbon and fiberglass and internal components of fiberglass. The testing was carried out by mounting the wing in an upside-down position on a composite beam bolted to a heavy steel table supported by a wooden post and a single strain rosette was attached to the strain indicators. The static test suggested a high amount of stress on the right skin for both horizontal and vertical directions; high stress was also found in the vertical direction of spar and rib which overall indicates that the average deflection mostly occurred at the wing tip. Kanesan et al. [4] carried out the experiment of static structural analysis on the UAV wing of NACA4415 airfoil with most components made of carbon fiber and Kevlar veil. The bending experiment was set up by applying loads on the bottom skin of the wing directly on top of main spar and aft spar with respect to area for each loading section. From the experimental results, higher deflection occurred near the wing tip relative to that from the wing centerline. Ramos [7] performed the static structural and tensile test experiment on the prototype of a long endurance electric UAV wing. The tensile tests were conducted for the internal and external parts of the wing for material of balsa wood, and fiberglass composite. The static test was conducted on the prototype before and after the heat-shrink film was applied, by applying four loads on the bottom surface of the spar. The result showed only small differences of displacement between with and without heat shrink film. The maximum deformation of the composite and balsa structure was found at the root of the wing, which is subjected to the elastic properties of the materials used.

The overall studies on both computation and experiment on the composite wing structural analysis are comprehensively explained in Table 1.

In Table 1, the findings highlight the current knowledge gap with a lack of works in material characterization especially for composite materials before being applied to the actual wing design. It can be observed that the maximum of two types of composite materials was assigned generally to numbers of the layer of the composite wing. The assignation of the multi-layered composite structure may have contributed to the strength and stiffness of the UAV wing. Besides that, a less recent experiment on a UAV wing was conducted applying the use of composite materials. Although it is relatively easy to replicate, it may consume more time in preparing the experimental setup and materials for the wing. This includes the limitation of cost to manufacture and test the wing and its structural parts. Therefore, wing structural analysis consists of optimizing several parts of wings such as skin, spar, ribs and stringer, including the connecting mechanisms which lead to the time and cost for conducting the experiments. The limitation of information has also been observed in the setting of computational simulation when correlating the fundamental of the laminated composite modeling with the application of FE analysis. Moreover, no study on the failure mechanism of the composite wing structure is available that highlighted the strength and stiffness of assigned laminated composite materials on the actual UAV wing.

Therefore, the simulation combining both interaction from structural aspects and composite characteristics that develops as useful tools for determining the structural response and improving the design of related structural-material stiffness. On top of that, in this research, the main methodology comprises three main stages, namely experimental validation, FE modeling with simulation validation and FE analysis on the multi-layered composite of UAV wing. The result of the analyses and discussion will be explained in Section 3. A conclusion will be provided in Section 4.

## 2. Methodology

In this study, the initial step of structural analysis is the experimental validation of composite laminates through experiments of tensile and flexural loads. In this step, the fabrication process and experimental tests of both tensile and flexural are prepared to study the behavior of composite laminates of the wing skin. The modeling process of FE simulation replicating the real domains of the experiment is the subsequent step of FE modeling and simulation validation. The analysis comparing both methods is carried out to prove the acceptance of using the simulation method. Further FE analysis on the actual size of composite UAV wing is conducted, by updating all the information on the setting of computational simulation correlating the fundamental of the laminated composite modeling. The overall process is presented in Figure 1.

### 2.1. Stage 1: Experimental Validation

The commencement step of the first stage is the fabrication of composite layered specimen for tensile and flexural experiments. Materials used in this experiment were carbon fiber (plain unidirectional fabric), Kevlar (plain-woven fabric) and Nomex honeycomb (aramid core with 2 mm height) which was supplied by ZKK Sdn. Bhd., Bandar Baru Bangi, Malaysia. The Nomex honeycomb is commonly used in aircraft wing flaps and rudders due to its high specific stiffness and low weight. The mechanical characteristics of materials of wing skin are referred to elsewhere [13]. The physical characteristics of materials of a specimen size of 300 mm × 300 mm are shown in Table 2. The fabrication process is depicted in Figure 2.

In detail, the composite plates of the part of wing skin were fabricated by stacking several pre-preg layers utilizing a hand layup process. For fabrication, the amount of matrix (resin) to be applied onto the fiber is dependent on the weight of the fibers. The amounts of fiber and matrix required were weighed and calculated referring to the fiber volume ratio from weight fraction [14].

In order to fabricate a square laminated plate, the Nomex honeycomb stacking between the fabric of carbon fiber and Kevlar was cut into 300 mm × 300 mm sheets. A square mold of 300 mm × 300 mm was sprayed with silicone spray to obtain a smooth sample surface and also to prevent adhesion. The layup started with the carbon fiber fabric, followed by Kevlar, core with Nomex honeycomb, Kevlar and carbon fiber fabric. The stacks of five layers of different laminates were placed in the center between two stainless-steel square molds and cold pressed for 5 min under 60 kg load. The specimens were cured for 24 h prior to post-curing at 80 °C for 2 h [15].

Hence, the fabricated plate of the hybrid laminated composite was cut into the required dimension for the purpose of tensile and flexural experiments.

#### 2.1.1. Experiment 1: Tensile Test

Rectangular specimens of 250 mm × 25 mm in length and width were cut from the laminated composite plate to conduct a static tensile test. This dimension was selected according to the ASTM D 3039 standard test for tensile properties of polymer matrix composite materials [16]. All five specimens were given with rectangular tabs at both end grips to avoid stress concentration. The experimental tests of tensile strength were run in displacement control by using an electro-mechanical testing machine (Bluehill Universal, Norwood, MA, USA) of INSTRON with a 10 kN loading cell. The crosshead speed was set to 2 mm/min and strain rate of 0.01/min. The strain was enforced within 0.5 degrees of fiber direction and measured from the beginning of the tensile test until the specimen break. An axial extensometer is used for strain evaluation.

#### 2.1.2. Experiment 2: Flexural Test

For performing the flexural test, the laminated composite plate was cut into a rectangular specimen. The dimension of 125 mm × 12.7 mm in length and width was based on Procedure A of ASTM D 790-03 standard test for flexural properties of unreinforced plastics material. It is also known as three-point bending tests that were conducted using a Universal Testing Machine (INSTRON 3382) (Bluehill Universal, Norwood, MA, USA with 10kN loading cell [17]. The distance between the two support spans (L) was 50 mm, with a cross-head speed of 1.33 mm/min.

### 2.2. Stage 2: Finite Element Modeling and Simulation Validation

In this stage, the most important step is the construction of laminated composite modeling according to the Classical Laminate Theory (CLT). By using this theory, the behavior of composite sandwich structure subjected to the external influence of load applied can be investigated. This case applies the Kirchoff–Love hypothesis of the CLT approach, in which the applied loads acting on the laminate structure are related to the mid-plane strains and curvature. By assuming both mid-plane and curvatures to be constant across the thickness, the stresses and strain in each lamina can be computed. However, by referring to [18,19], several assumptions were considered, as follows:Each lamina is orthotropic and quasi-homogeneous;Deformation is continuous and small through the laminate;The laminate is thin and is loaded in its plane (plane stress), whereby out-of-plane (normal) direct stress is zero;Layers are perfectly bonded together and no-slip occurs between the lamina interfaces;Transverse shear strains ( and $\gamma_{yz}$) are negligible;Transverse normal strain $\varepsilon_{z}$ is negligible compared to the in-plane strains $\varepsilon_{x}$ and $\varepsilon_{y}$.

In modeling the laminated composite using ANSYS, the geometry model of specimen size is interpreted in the form of a shell element (lamina). Firstly, the lamina needed to be defined according to fabric, material and its thickness. Then, the rosette as a reference direction of 0° represents the fiber direction, the oriented selection set for layup direction and composite layup in Modeling Ply Groups. In this simulation, each wing component has a different setting of the rosette (fiber direction) indicated by the position of the component subjected to the direction of XYZ axes, while the oriented selection set refers to layers that are laid on each other based on the vector of layup direction.

Theoretically, a laminate comprises the number of plies, *n* and each ply has a thicknesses of $t_{k}$. The thickness of the laminate, *h* can be derived as
(1)$$h=\sum_{k=1}^{n}t_{k}$$

The location of the mid-plane is $\frac{h}{2}$ from the top and the bottom surface of the laminate. The z-coordinates of each ply *k* surface are given by:

Ply 1:       (top surface)
$h_{0}=-\frac{h}{2}$
(2)
       (bottom surface)
$h_{1}=-\frac{h}{2}+t_{1}$
(3)Ply k:       *(k = 2,3,…..n−2, n−1)*
(4)
       (top surface)
$h_{k-1}=-\frac{h}{2}+\overset{k-1}{\underset{-1}{\sum }}t$
(5)
       (bottom surface)
$h_{k-1}=-\frac{h}{2}+\overset{k}{\underset{-1}{\sum }}t$
(6)Ply n:       (top surface)
$h_{n-1}=-\frac{h}{2}+t_{n}$
(7)
       (bottom surface)
$h_{n}=\frac{h}{2}$
(8)

The laminate created is depicted in Figure 3.

The construction of the composite components mostly used are carbon fiber fabric, Kevlar, honeycomb core with epoxy resin as the matrix.

#### 2.2.1. Tensile: Mesh Generation and Boundary Conditions

In ANSYS Workbench 17.0, after the construction of the laminated composite specimen as explained in Section 2.2, the shell element from ACP (Pre) domain was imported into implicit static structural analysis. The updated geometry required mesh generation. Then, the discretization of geometry, whereby the grid dependency tests of mesh elements were conducted. The mesh dependency check is carried out varying the number of elements and the best mesh of quadrilateral elements of 1625 is selected with good quality of 0.20. It should be noted that the maximum skewness of cell quality with the value towards 0 is indicated as the excellent cell quality with ideal and skewed quadrilaterals cell.

For boundary conditions of the tensile specimen, the first boundary condition of fixed support was assigned at the end grip with a tab of the rectangular specimen. While the second boundary condition was set to be towards the *Z*-axis with a force of 0 N. The other end of the rectangular specimen was assigned as the loading condition of load moving outward with a velocity of 20 mm/s until the specimen failed. The simulation ran for 10 s, with a time step of 0.1 s. The calculation was considered to be accomplished once the specimen failed. The simulation setup replicating the actual experiment for the tensile test is depicted in Figure 4.

#### 2.2.2. Flexural: Mesh Generation and Boundary Conditions

Meanwhile, for flexural testing, the similar process was used of importing the updated geometry from ACP(Pre) domain to the explicit dynamic of ANSYS Workbench 17.0. The explicit dynamic is required in conducting the flexural test due to its requirement of transient dynamic forces to determine accurate material deformation with high strain rates, which is difficult to solve with implicit solution methods [20]. Similar to tensile tests, the grid dependency tests of mesh elements were carried out, and hence the best element of the quadrilateral cell of 2490 is selected with a fair quality of 0.53.

For flexural testing, the boundary condition of fixed support was assigned on the two supports as both sides of the bottom of the rectangular specimen, which was according to its required standard for flexural tests. The second boundary condition was set at the bottom part of the specimen, with downward displacement on the *Y*-axis of −15 mm. For this simulation, the loading condition was set and applied at the center part of the rectangular specimen. In particular, the loading nose was applied by moving downward with the velocity of 10 m/s until the specimen failed. The simulation ran for 2 s, and the step end time was set to be 0.05. In this domain, the material was assigned as a failure. The simulation setup replicating the actual experiment for the flexural test is depicted in Figure 5.

Both simulations were performed and compared with data from experiments. Thus, the acceptable results obtained from both analyses of tensile and flexural properties may progress to further FE analysis on an actual UAV wing by updating the similar process parameter settings of simulations.

### 2.3. Stage 3: Finite Element Analysis on UAV Wing

The main element of this stage is the updating of process parameter setting on the geometric design of the UAV wing. A similar procedure setup is assigned that basically starts from the modeling of external and internal parts of the wing, meshing, boundary and loading conditions, laminating the geometry with composite materials before performing the FE-ACP static structural simulation.

#### 2.3.1. Geometrical Modeling of UAV Wing

With the aid of Computer-Aided Design (CAD) software of SolidWorks 2013 (Dassault Systemes SolidWorks Corporation, Waltham, MA, USA), the ALUDRA MK-1 wing was designed. In this study, the specification of design has important constraints such that the airfoil wing of NACA 4415 and its wingspan have been predetermined. The important parameters of the ALUDRA MK-1 UAV wing are tabulated in Table 3.

The shape of the airfoil is the most important aspect to design the internal components of the wing. The wing model with tubercles at the leading edge (TLE) is presented in Figure 6a. In a similar study by [13,21], the spherical pattern of tubercles at the leading edge of the TLE wing was designed by using optimum amplitude, A of 0.025 c and wavelength, λ of 0.25 c due to its remarkable effect on the airfoil performance [22,23,24]. The configuration of the TLE wing is depicted in Figure 6b. The configuration of the internal components of the TLE wing is in Figure 6c.

#### 2.3.2. Updating: Mesh Generation, Boundary and Loading Conditions

In this study, the wing is categorized under a thin-wall structure due to the composite materials used; thus, the 3D shell element is suitable for constructing the mesh of the wing structure. The quadrilateral (square) cells are used to mesh the geometry with simple regions such as wing skin (normal) and spars. However, the tetrahedral grid with connecting spacing is the preferred element size to mesh the complex region of the wing geometry such as ribs and TLE wing skin. Before generating the mesh, it is crucial to ensure that the mesh should accurately represent the geometry of the computational domain and load. The mesh also should adequately represent the stress gradient or large displacement in the solution. The grid independence test is carried out by the varying number of elements to determine optimized mesh density, as in Figure 7. From the dependency test, the 250 k element mesh provided an accurate solution with the quadrilateral mesh. The maximum skewness of 0.8 is considered as good mesh cell quality as the value is close to 0.

On the other hand, the boundary condition for the wing model is set from a known value for a displacement or an associated load [1]. In this context, the boundary conditions are based on DOF constraint at specified model boundaries for defining the rigid support points. The displacement or force DOF at particular nodal points of the wing model is specific for certain cases. In this analysis, two important boundary conditions needed to be assigned. The first is the bracket attached to the fuselage. The bottom surface of the bracket is defined as fixed by using a screw and nut due to the fuselage not being included in the model. Hence, this defines that all the rotations and displacements of the bottom surface of the brackets are set to zero, UX=UY=UZ=ROTX=ROTY=ROTZ=0 [13], [21]. The second boundary condition is the symmetrical condition of the wings, which considering that half of the wing that developed in FE analysis. The spars of the wing are set as symmetrical about the XZ-plane. Hence, the rotation along *Y*-axis and *Z*-axis, as well as the displacement along *X*-axis were assigned as zero, UX=ROTY=ROTZ=0 [13,21].

Moreover, the wing loading condition is the most critical aspect in the design of the wing. In this study, the loads applied the spanwise lift distribution of Schrenk’s method. The calculations of the loads on the wing require considering the lift forces generated by the wing that are distributed along the wingspan. The most aerodynamically efficient and practical wing will have an elliptical distribution of lift along the span, where zero lift is generated at the tip and maximum lift is generated at the wing centerline. This assumption is supported by [25], where 80% of lift load acted on the wings and the remaining 20% acted on the fuselage. Thus, the maximum load is acting nearer to the wing roots. In this study, the aerodynamic loading used a simple yet practical method to solve spanwise lift distribution, which can be referred to in [26].

#### 2.3.3. Updating: Lamination Process and Composite Modeling

Using a similar process as prepared for the validation process, the laminated composite is defining mechanical properties, base materials as fabric, ply type and additional failure criteria. The notation and ply arrangement in the laminate are presented by considering the shell elements to model the wing structure. The modeling process for laminating composites is referred to as the construction of laminated composite materials [13]. In ACP (PrePost) module, the wing model is developed in the form of a shell element of SHELL181 that is assigned as a single sheet according to its thickness. Then, the composite fabrics can be created along with other composite layers and core materials, which are subsequently combined to form the laminated composite materials. By creating the composite fabric based on its thicknesses, a stack-up can be created by attaching all fabrics according to the assigned orientation (°). Then, a laminate is created, the stack-up with an orientation of 0° is sandwiched between the top and bottom plies. One of the most challenging parts is the complication during design configuration when combining various materials with different plies and various orientations in a 360° manner. This is crucial to adequately assign the model configuration related to local coordinate systems (x^1^, y^1^, z^1^) of an assigned orientation, considering the other plate stacking off-axis plies according to its appropriate local coordinate system (x^3^, y^3^, z^3^) [19].

Referring to Equations (1)–(8), the material orientation is acquired based on geometric attributes, which include the layup direction, layup area and reference direction. In this simulation setting, the rosette is defined according to the fiber direction. The coordinate direction is defined as how the materials are oriented. Then, the oriented selection set is subjected to layup direction. In this case, the layup area of element sets is selected to define the reference direction and its orientation directions. Finally, the local and global plies are defined to generate a new modeling ply of laminated composite, replicating the fabrics laid onto a mold. The updated assignment of laminated materials on the structural components of the wing was presented in terms of material layup perspective and material orientation perspective, respectively, as in Figure 8.

The updated assignment of laminated materials on the structural components of the wing is presented in terms of material layup and material orientation perspectives, respectively.

#### 2.3.4. Updating: FE-ACP Structural Simulation

In regard to the numerical approach of integrating the composite aspect of ACP (PrePost) and FE analysis of static structural analysis, the intuitive modeling of composite layup using simulation software is a prerequisite for the modeling process following manufacturing; hence, the simple and fast modification of the layering and design of composites is able to be accomplished during the FE analysis. The integration of both FE static structural and ACP (PrePost) was connected systemically by transferring the generated solid element model from ACP (Pre) to the static structural to further analyze and evaluate the composite design in ACP (Post). Hence, the result obtained from the FE-ACP analysis in static structural analysis was evaluated from composite perspectives, which were presented in the post-processing of ACP(Post).

## 3. Post-Processing of FE-ACP Simulation: Failure Analysis

According to Sun et al. [27], the failure criteria of a lamina can be divided into three groups—limit criteria, interactive criteria and separate mode criteria. The limit criteria involved the prediction of failure load and mode through the comparison of the stress of lamina (σ_11_, σ_22_, τ_12_) subjected to strength separately, but without considering the interaction among the stresses. Most importantly, it should be noted that all the composite materials used in this study are classified as orthotropic materials due to each material differ along three mutually orthogonal axes. According to Cazacu [28] and Cogun et al. [29], the orthotropic yield criteria can be demonstrated which are expressed by a quadratic yield function under plane stress conditions. In the ACP domain, all the stress criteria were defined according to the required direction, which are known as in-plane stresses of longitudinal (*s1*), transverse (*s2*) and shear (*s12*) as well as principal direction (*sI*), whereas the strain criteria are also defined by the required direction, known as in-plane strains of longitudinal (*e1*), transverse (*e2*) and shear (*e12*) as well as principal direction (*eI*).

Meanwhile, the interactive criterion is the prediction of the failure load of stress or stain using a single quadratic or higher-order polynomial equation. The stress/strength ratios are compared for assessing the failure mode, which considered the failure of Tsai–Wu.

The Tsai–Wu criterion is applied in order to examine the existence of a failure surface between the tensile strength and compressive stress of a lamina [30]. The calculation and reduction of the Tsai–Wu criterion for a plane state of stress can be written as [31]:(9)$$F_{1}\sigma_{1}+F_{2}\sigma_{2}+F_{11}{\left(\sigma_{1}\right)}^{2}+F_{22}{\left(\sigma_{2}\right)}^{2}+F_{66}{\left(\tau_{12}\right)}^{2}+2F_{12}\sigma_{1}\sigma_{2}\geq 1$$
where $\sigma_{1}$ is stress in the fibre direction, $\sigma_{2}$ is stress in transverse direction and $\tau_{12}$ is shear stress. This equation can be used to assess failure in a composite lamina and the expression of coefficients for longitudinal strength, transverse strength and shear strength [31]. Hence, the interaction coefficient can be expressed a follows [31]:(10)$$F_{12}≅-\frac{1}{2}{\left(F_{11}F_{22}\right)}^{1/2}$$

In this failure criterion, the tensile and compressive stresses can be distinguished due to significant coefficients. It also can be easily incorporated into automated computational procedures.

Most importantly, the margin of safety (MoS) is related to the Tsai–Wu criterion, which expressed as:(11)MoS=SR−1
where SR is the strength ratio. It can be obtained from the following equation:(12)$$SR_{TW}=\frac{\left(-b+\sqrt{b^{2}+4a}\right)}{2a}$$
where a and b are subjected to the following parameters:(13)$$a=F_{11}{\left(\sigma_{1}\right)}^{2}+F_{22}{\left(\sigma_{2}\right)}^{2}+F_{66}{\left(\tau_{12}\right)}^{2}+2F_{12}\sigma_{1}\sigma_{2}$$
(14)$$b=F_{1}\sigma_{1}+F_{2}\sigma_{2}$$

In this simulation, the failure plot can be applied to present the safety factor that is defined as the reserve factor. The critical layer index is counted from the reference surface upwards and denoted as layer 1. The critical load case index is started at 0.

## 4. Results and Discussion

### 4.1. Validation: Experiment vs. FE-ACP Simulation

#### 4.1.1. Tensile Test

In this experiment, five specimens were tested for tensile strength until failure. Considering mean and standard deviation, it can be realized that the tensile stress values are close to one another. Hence, the result of tensile stress test subjected to the strain comparing both experiment and simulation is shown in Table 4 and Figure 9.

As in Table 4, the tensile experiment showed higher tensile strength and tensile modulus with 257.46 ± 20.22 MPa and 17.39 ± 1.1 MPa, whereas tensile simulation displayed a slightly different value in both strength and modulus with 228.86 ± 20.22 MPa and 15.89 ± 1.1 MPa with 11.1%. It can be noticed that the error bar value for tensile strength of both simulation and experiment showed a quite high value due to various factors of human error during the fabrication such as the spreading of resin onto each laminate and the high number of layers of composite laminates [32].

Referring to Figure 9, it was observed that the simulation result produced an almost similar pattern of tensile load at tensile displacement. For the experiment, the hybrid laminated composite showed a slight curve at the early phase of tensile testing due to the presence of Kevlar material. The graph of simulation demonstrated a linear stress–strain relationship prior to achieving maximum point after 0.6% strain. From both experiment and simulation, it can be noted that the material behavior of Kevlar has a higher tendency of elongation before the break due to its properties of high strength and high modulus. Carbon and epoxy have brittle properties compared to Kevlar. Furthermore, honeycomb also reaches critical stress and fails due to brittle crushing under a high enough tensile load. Quan et al. [33] proved the honeycomb behavior under in-plane tensile load along the longitudinal direction, which showed a sudden drop after reaching peak stress due to instability or defects forming layer by layer in an asymmetric way.

#### 4.1.2. Flexural Test

In the second experiment, five specimens were tested for flexural properties until failure. By taking into account the mean and standard deviation, the values of the flexural load were observed to be close to one another. From the physical condition, both results of both experiment and simulation of the flexural specimen are depicted in Figure 10a,b.

Referring to Figure 10a, the specimen showed that the highest deformation occurred in the middle part of the specimen subjected to the applied load moving downward. The material was not broken and only showed damage that occurred in the middle part. This was mainly due to the properties of Kevlar compared to other materials. A similar condition was observed on the actual specimen, as in Figure 10b. It was found by Dong et al. [34] that the theoretical results for the flexural experiment were generally greater than the simulation results. This may be due to the existence of shear stress in the test specimen, hence resulting in additional deformation. Hence, the flexural result of load–time in the graph is shown in Figure 10c; it was observed that the simulation result produced an almost similar pattern of flexural load versus time. The load–time response showed a 10.47% difference of flexural simulation less than flexural experiment data.

From the experiment, the purpose of conducting both tests of flexural and tensile properties is mainly to validate the numerical modeling used in the simulation and correlate the process parameter setting of the simulation which replicates the actual experiment tests. Therefore, according to the good agreement between the results obtained, the process parameter setting for simulation is acceptable for further analysis on the actual design of wing skin.

### 4.2. FE-ACP Simulation on TLE Wing

#### 4.2.1. Result: Total Deformation

The results of contour plots indicated the variation of deformation in the form of colors. The red color is defined as the critical value, whereas the dark blue color is defined as the safest value. In fact, the distribution of deformation on the overall wing is determined according to the maximum point at the tip of the wing from the wing root that is connected to the fuselage. The result of total deformation for TLE wing under aerodynamic loading is shown in Figure 11.

As in Figure 11, the current FE-ACP simulation showed the resulting total displacement of 33.916 mm, where the maximum value occurred at the tip of the wing on the wing skin. Due to the combination of orthotropic materials, the wing has proven to be much stronger as extra loads are applied to it. Yet, the value of the total deformation obtained was almost ten times higher than the second parametric study. The result of deformation is feasible and acceptable for aerodynamic and structural conditions. Qualitatively, the distribution of skin deformation was slightly tilted at the trailing edge near to the root of the wing but was uniformly distributed as approaching the tip of the wing. Meanwhile, the spar displacement was slightly less than the skin with 33.896 mm, the maximum at the outboard rib of the wingtip.

#### 4.2.2. Result: Stresses

From the perspective of material characteristics, the results of stresses were determined based on the directions of in-plane and principal. The criticality of the stresses was explained according to the legend colors, in terms of qualitative and quantitative perspectives.

##### In-plane Stresses

In this case study, the results of in-plane stresses were computed to understand the significant stresses behavior under aerodynamic loading conditions. The total in-plane stresses are shown in Figure 12a–c, as well as the details of strains in Table 5.

From observation, the *s1* (longitudinal direction) of the TLE wing showed total stress of 197.9 MPa. As in Figure 12a, the stress distribution of *s1* was dispersed for the whole bottom part of the wing, whereby the green color showed the negative value of *s1* (as referred to as legend color) for the area of trailing edge and the area nearest to the wing root and tip with a positive value for the most of area of the bottom side of the wing (light green in color). As referred to in Table 5, the highest *s1* was on the spar and less on the skin with the value of 80.981 MPa. Meanwhile, for *s2* (transverse direction), the stress obtained is slightly less than *s1*, with a total value of 194.260 MPa. As in Figure 12b, the stress *s2* was distributed mostly around the area nearest to the root of the wing until the middle side of the bottom wing. The stress of *s2* mostly occurred on the spar, yet skin showed a lower value of 99.002 MPa. For *s12*, 31.129 MPa was obtained for the total stress with the stress mostly distributed on the main spar nearest to the tubercles pattern at the wing root. Similarly, the spar showed the highest, while skin only had a value of 10.190 MPa. Hence, it can be noted that under the real condition of aerodynamic loading, the stress concentration mostly occurred on the internal components of spars. This is due to the design of the spar with the shell frame affected by the localized increase in stress. As stated by Anderson [35], the extremely high amount of loading can be transferred and distributed from the skin to other internal structures until impact deformation occurred. Hence, the strength and stiffness of the structure of the TLE wing are practically optimized through the best configuration of the wing with efficient use of material.

##### Principal Stress

The other findings of principal stress of *sI* for the TLE wing are shown in Figure 13 and Table 6.

Referring to Figure 13, the highest value of *sI* was found at the main spar with the value of 199.04 MPa. Hence, the distribution of total stress *sI* was between 39.807 MPa to 89.566 MPa located on the skin nearest to both spars. As referred to in Table 6, skin showed a lower value of maximum stress of *sI* with 100 MPa, compared to spar.

#### 4.2.3. Result: Failure Criteria

Subsequently, the most essential finding in this FE-ACP simulation is the failure criteria that are influenced by the criticality of loading subjected to wing structural failure. The prediction of failure load and mode was mainly subjected to the interaction among the stresses for Tsai–Wu.

In this result of failure criteria, the assessment of failure was calculated considering the stresses in the direction of longitudinal (also known as fiber direction, (*s1*), transverse direction (*s2*) and shear stress (*s12*). Thus, for further understanding, the calculated stresses contributed to the computed Tsai–Wu failure due to the interaction among stresses on the wing. The failure criterion of Tsai–Wu is depicted in Figure 14a. Referring to Figure 14a, the area of Tsai–Wu failure criterion was nearer to the spar of the wing.

Hence, the maximum stress and strain were obtained from the post-processing in the ACP domain, as depicted in Figure 14b. As in Figure 14b, the maximum stress was captured from the upper part of the wing located at the main spar. The critical spar was found with the red color, which is denoted as *s2c(7)*, and indicated that the transverse stress with compressive stress occurred on the spar at the seventh layer, while at the skin, the stress of *s1c(2)* displayed that the longitudinal direction of stress *s1* occurred compressively on the second layer of the skin.

The failure elements were expressed in terms of reserve factor, according to the legend colors. Hence, the zoomed figure of the affected area is shown in Figure 14c,d. Referring to Figure 14c, the Tsai–Wu failure was found on the main spar of the monocoque-foam-reinforced TLE wing under aerodynamic loading. The upper part shows that the element of spar failed at *tw(8)* and *tw(7)*. Thus, this condition denoted that the spar failed at the eighth and seventh layers, with an RF value of 1.071 and 1.000, which exceed the value of 1.000. The Tsai–Wu failure was noted to show a noteworthy effect at transverse stress state. Hence, failure can predict lower laminate strengths [36]. Similar to Figure 14d, the red color indicates that the aft spar failed at the critical seventh layer with an RF value of 1.071. The other colors indicate the elements still considered as not failed due to the RF value obtained within the limit and not exceeding 1.000.

Overall, incorporating the theories of laminated composites and numerical simulations of FE is highlighted in this study to understand the structural response and improve the design of related structural-material stiffness. In addition, relevant decisions related to the aero-structural performance of the wing can be made to improve on today’s modern requirement for UAV. With the aid of numerical simulation of FE-ACP, the aero-structural performance of structural-materials combination can be helpful and applicable to industries, especially aircraft engineering and manufacturing.

## 5. Conclusions

From FE-ACP simulation, the total deformation is one of the parameters used to define structural performance. The finding is the realistic and practical results of deformation that are subjected to calculated aerodynamic loading. Hence, Schrenk’s method can be used for applying external loads in FE-ACP simulation. The material characteristic of stresses and strains were considered to be subjective depending on the reinforced structural configuration. The reason is the stress and strains were mainly due to the stress concentration on the overall wing components including both skin and spars. The stress concentration was mainly affected by the presence of tubercles at the leading edge of the wing. Therefore, the use of stresses in a particular direction is mainly to understand the characteristic of stress when the load is applied. Interestingly, in composites, stress concentration and Poisson effect influenced the behavior of stress and strain especially in the in-plane directions and principal direction. As the TLE wing was subjected to the aerodynamic loading of lift distribution by Schrenk’s, the outcome from this FE-ACP analysis is the prediction of failure mode under interaction among stresses and strains. From Tsai–Wu failure criterion, the monocoque-foam-reinforced TLE wing showed a unique failure mode and behavior according to the interaction among all computed stresses such as *s1*, *s2* and *s12*. According to the observations, the failure of Tsai–Wu was mostly found on the spars of the wing and skin nearly to the root of the wing. The critical failure mode of Tsai–Wu was noted mostly at the seventh and eighth layers of both spars.

## Figures and Tables

**Figure 1 materials-14-03705-f001:**
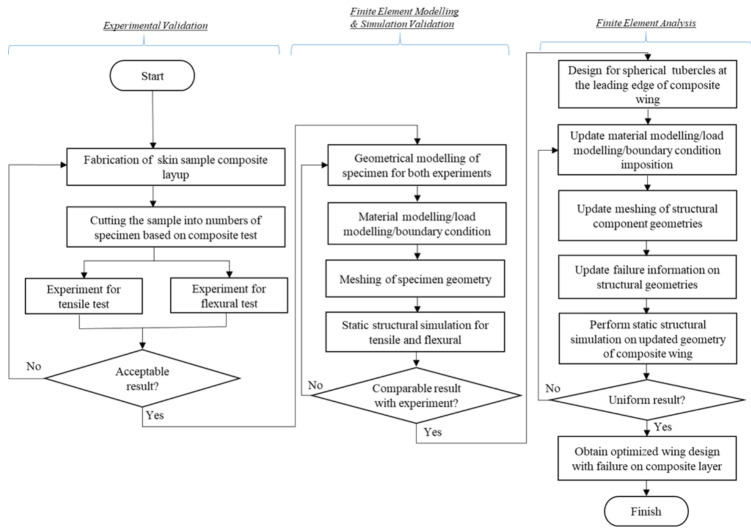
Methodology of multi-layered composite lamination for structural analysis.

**Figure 2 materials-14-03705-f002:**
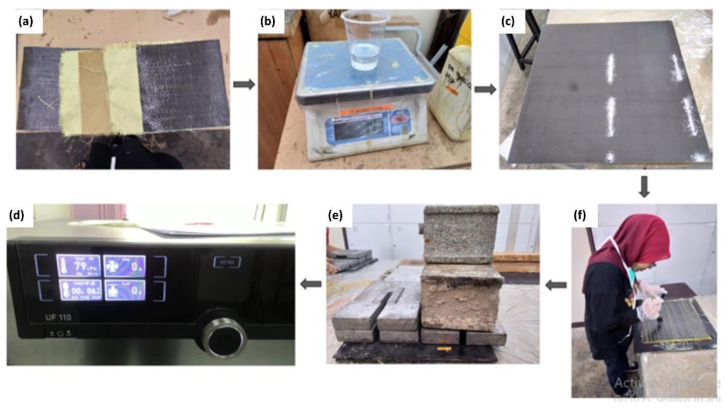
Fabrication process (**a**) Stacking of composite layers (CF, KV, HV, KV, CF) (**b**) Weighting the amounts of epoxy resin and hardener (**c**) Applying silicone spray on the square mold (**d**) Layering up the composite layers and applying the resin in between the layers (**e**) Cold-pressing the laminated specimen for 5 min under 60 kg load (**f**) Curing the laminated specimens at 80 °C for 2 h.

**Figure 3 materials-14-03705-f003:**
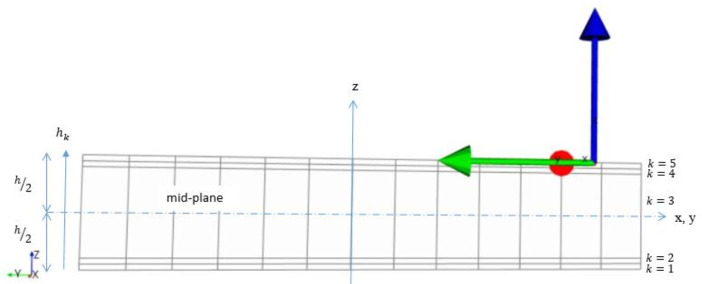
Lamination arrangement of composite in ANSYS module.

**Figure 4 materials-14-03705-f004:**
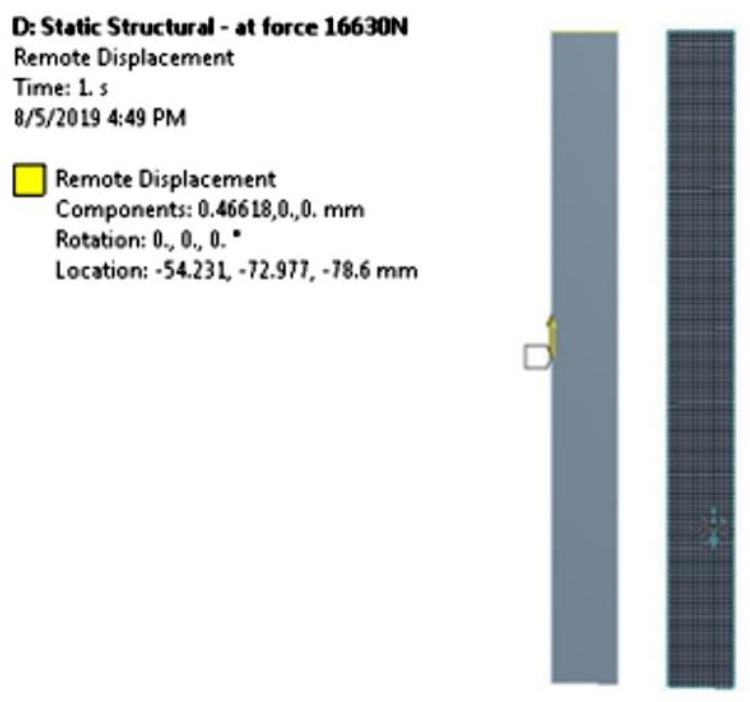
Simulation setup for tensile test.

**Figure 5 materials-14-03705-f005:**
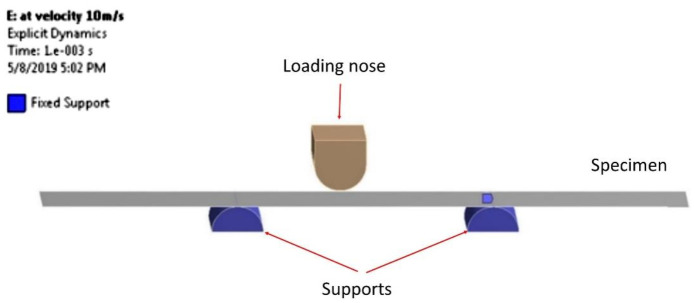
Simulation setup for flexural test.

**Figure 6 materials-14-03705-f006:**
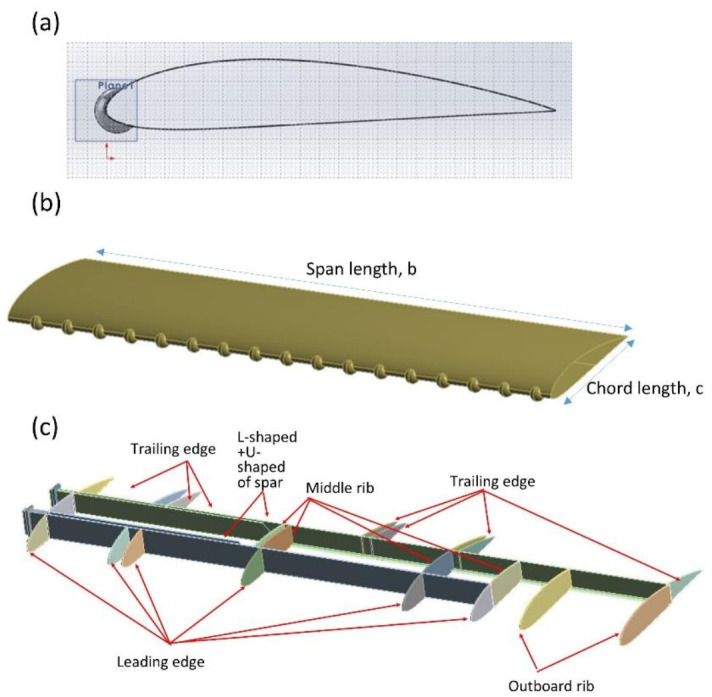
(**a**) NACA4415 airfoil for TLE wing; (**b**) configuration of TLE wing skin; (**c**) configuration of internal components of TLE wing.

**Figure 7 materials-14-03705-f007:**
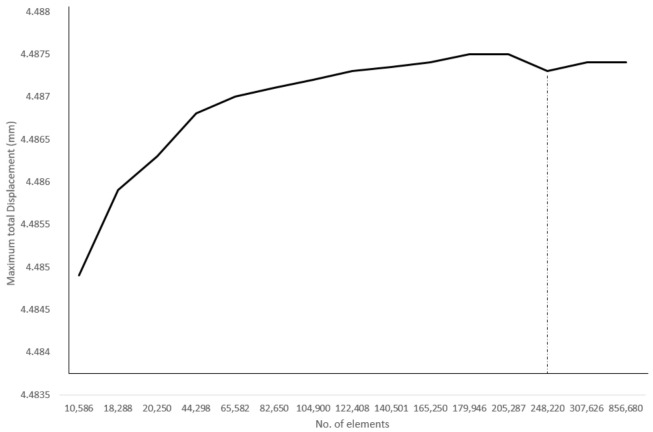
Grid dependency test subjected to total displacement.

**Figure 8 materials-14-03705-f008:**
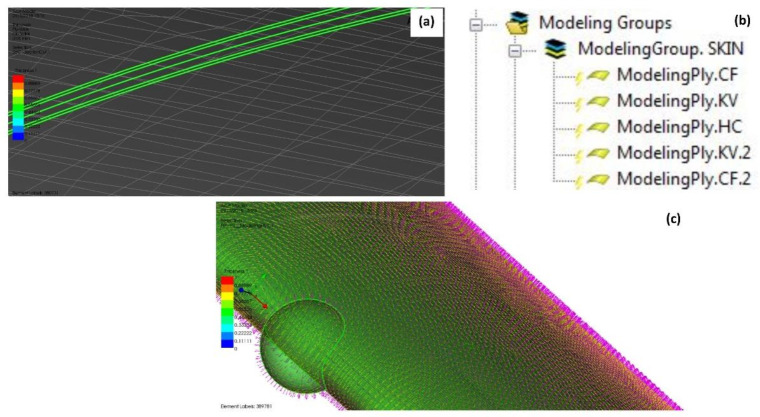
(**a**) Visualize of lamination of materials; (**b**) updated modeling groups according to lamination of the materials; (**c**) orientation direction of composite (red color), reference direction of elements (yellow color) and transverse direction of layer (green color).

**Figure 9 materials-14-03705-f009:**
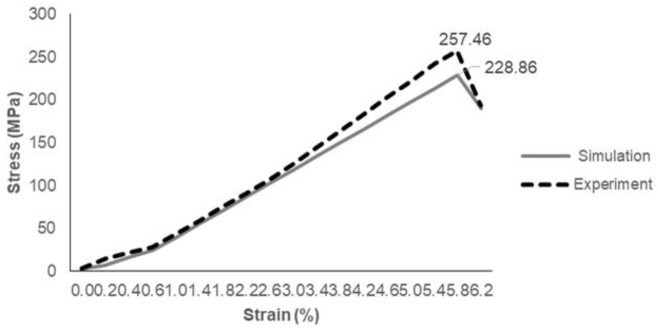
Tensile stress versus strain of experiment and simulation data.

**Figure 10 materials-14-03705-f010:**
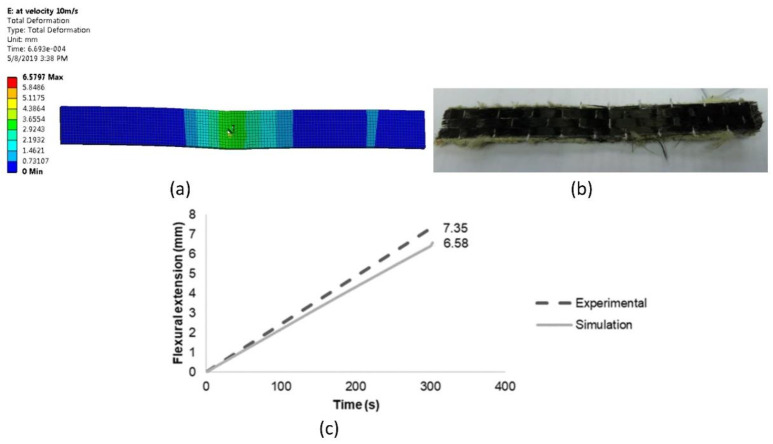
(**a**) Flexural ACP simulation; (**b**) flexural experimental specimen; (**c**) flexural extension versus time of experiment and simulation data.

**Figure 11 materials-14-03705-f011:**
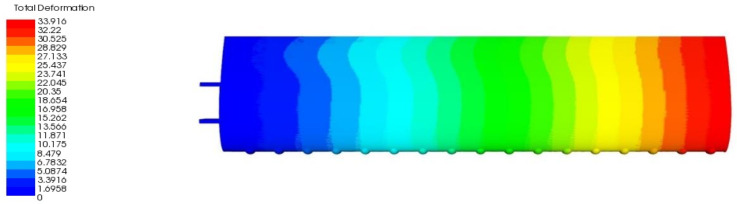
Result of total deformation for TLE wing under aerodynamic loading.

**Figure 12 materials-14-03705-f012:**
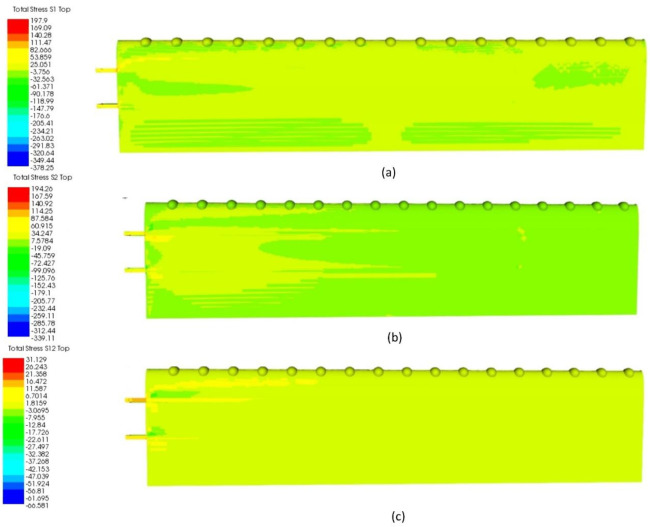
(**a**) In-plane stress (*s1*) for TLE wing under aerodynamic load; (**b**) in-plane stress (*s2*) for TLE wing under aerodynamic load; (**c**) in-plane stress (*s12*) for TLE under aerodynamic load.

**Figure 13 materials-14-03705-f013:**
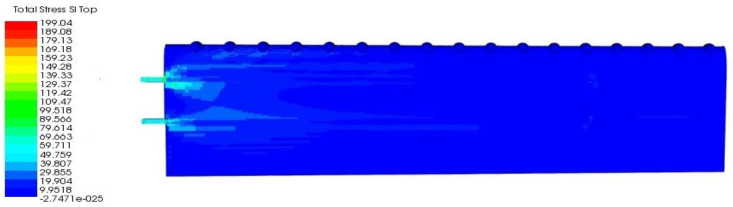
Principal stress (*sI*) for TLE wing under aerodynamic load.

**Figure 14 materials-14-03705-f014:**
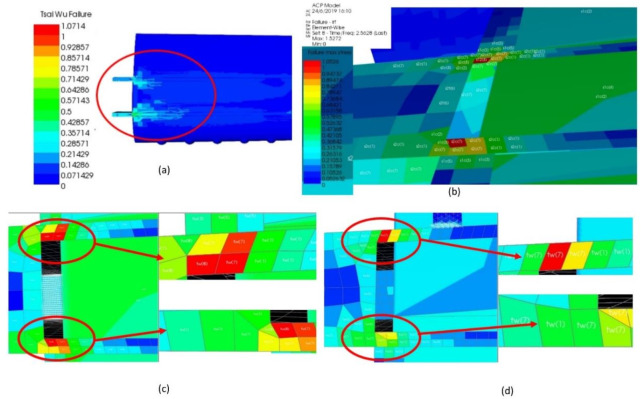
(**a**) Tsai–Wu failure for TLE wing under aerodynamic load; (**b**) maximum stress at spar of TLE wing; (**c**) Tsai–Wu failure at main spar; (**d**) Tsai–Wu failure at aft spar.

**Table 1 materials-14-03705-t001:** Computational and experiment methods for UAV structural analyses.

Software	Wing Type/Airfoil	Analysis	Wing Section	Material Properties	Method	Ref.	Gap
ANSYS	NACA4318	Static linear	Skin	Vectran (urethane-coated)	Computational/ Experiment	[8]	The structure of skin is inflatable, and the analyses do not consider the internal components as it may affect the deformation of the wing
		Static nonlinear	Skin	Vectran (hyperelastic material)			
	NACA0012 (taper) and NACA2412 (rectangular)	Static (Orientation and material composition)	Skin	Graphite/ epoxy	Computational	[5]	The wing skin used composite material but the other wing components used aluminum. The best composition resulted in less deformation but high in weight.
			Spar, ribs, flanges and stringer	Aluminium 7075-T6			
	UAV	Static linear, Dynamic, Buckling static	Skin with/without spar and rib	S Glass, Kevlar, Boron Fiber	Computational	[6]	The analysis only for the wing skin and the load is only air pressure, need to consider loads act on spar
	UAV/ Jedelsky profile	Static and Dynamic	Skin	Glass fiber, foam core and	Computational/ Experiment	[11]	No spars or ribs, the skin is only thin curved plate attached with actuators.
			Actuator	Micro fiber			
PATRAN	NASA/LANGLEY LS (1) (GA (W)-1)	Static	Skin	Carbon fiber reinforce plastic composite	Computational	[2]	The maximum stress only observed at root of the wing not the entire wing
	UAV	Static linear and Static linear various orientation	Spars	Aluminium 7075-T651	Computational	[3]	The structural optimization of the wing only assumed the loads acted on the wing to be 500N
			Ribs and fittings	Aluminium 2024-T3			
			Skin	Graphite/ epoxy			
ABAQUS	Ultralight UAV	Static nonlinear	Skin	Foam core	Computational/ Experiment	[9]	Significant deviation on the stacking ply sequence of composite skin due to adhesive thickness of resin and placement of strain gauge
			Spars, Ribs, Spars interconnectors	Carbon fiber/ epoxy			
	NACA4415	Static	Skin, Ribs	Carbon fibre fabric, Kevlar veil	Computational/ Experiment	[4]	The analyses are in good agreement for composites materials on the whole wing design. The materials can be used as referenced for further studies on composite
			Spar	Carbon fibre fabric and unidirectional carbon fibre			
Pro-Engineer	NACA2415	Static	Skin	Carbon with fiberglass	Computational/ Experiment	[12]	No specific load and no stresses on the skin and rib are observed for the simulation due to simulation inadequacies
			Spars, Ribs	Fiberglass			
CATIA Generative Structural Analysis	Long Endurance Electric UAV	Static linear	Spars, Ribs	Balsa wood	Computational/ Experiment	[7]	The analyses found that 12% difference between experiment and computational analyses, thus caused a huge difference for the study of the wing
			Reinforcement	Fiberglass			
			Boom	Pultruded carbon			
			Skin	Plastic			

**Table 2 materials-14-03705-t002:** Physical properties of materials of specimen size of 300 mm × 300 mm (Manufacturer datasheet).

Material	Thickness (cm)	Weight (g)	Volume (cm^3^)	Density (g/cm^3^)
Carbon fiber (CF)	2.7	40	24.3	1.646
Kevlar veil (KV)	2.5	43	22.5	1.912
Nomex honeycomb (HC)	20	7	180	0.0389
Epoxy	–	–	–	1.14

**Table 3 materials-14-03705-t003:** Important parameters of the wing.

Parameter	Value
Span length, b (m)	5.1257
Chord length, c (m)	0.5886
Wing area, S (m^2^)	3.5295
Taper ratio	1.0
Aspect ratio, AR	10.53
Maximum load factor, n	3.8
Maximum weight, w_0_ (kg)	200

**Table 4 materials-14-03705-t004:** Tensile strength and tensile modulus of laminated specimen.

Method	Tensile Strength (MPa)	Tensile Modulus (MPa)
Experiment	257.46 ± 20.22	17.39 ± 1.1
Simulation	228.86 ± 20.22	15.89 ± 1.1

**Table 5 materials-14-03705-t005:** Details of in-plane stresses *s1*, *s2* and *s12* for TLE wing under aerodynamic load.

	Maximum Stress (MPa)	*s1*			*s2*			*s12*		
Wing Type		Total	Spar	Skin	Total	Spar	Skin	Total	Spar	Skin
TLE wing		197.90	197.90	80.98	194.26	194.26	99.00	31.13	31.13	10.19

**Table 6 materials-14-03705-t006:** Details of principal stress *sI* for TLE wing under aerodynamic load.

	Maximum Stress (MPa)	*s1*		
Wing Type		Total	Spar	Skin
TLE wing		199.040	199.040	100.000

## Data Availability

The data presented in a publicly accesible repository. The data presented in this study are openly available in FigShare at 10.6084/m9.figshare.14552985, reference number 14552985.

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
