# Peer review of "Consideration of Lamination Structural Analysis in a Multi-Layered Composite and Failure Analysis on Wing Design Application"

_materials, 2021, doi:10.3390/ma14133705_

Round 1

Reviewer 1 Report

The first fact that needs to be mentioned is the very poor English language in which this paper was written. To the extent that the work is incomprehensible in some parts. There are to many mistakes that it is impossible to mention them all.

The abbreviations should be explained in the first appearance.

In line 61, "Von Mises at the root of the wing" - I understand what Von Mises is suposed to be, but detailed expression should be given.

Line 249 - "the rosette" - detailed explanation? 

Section 2.2.1. Authors mention mesh quality of 0.20 and 0.53. Can you explain those numbers? Mesh quality values differ from software to software.

Sentence in line 294 to 297 are also very incomprehensible. In section 2.3.2, line 353, listing of the mesh sizes is pretty unnecessarily and opaque. A diagram would be better here. Figure 8. is also needless, in rank of student seminar work, not for serious scientific paper.

Can you explain the ±20.22 value in Table 4.? How did You obtain that specific (quite high) value? 

The general assessment, i.e. the scientific value of this paper is quite low, as this is a topic that has been discussed many times, even as an example in some student books and books on the finite element method.  I am not sure what the scientific effect, i.e. the contribution of this paper is.

Reviewer 2 Report

Reviewers comments on the paper

 Consideration of Lamination Structural Analysis in a Multi-layered Composite and Failure Analysis on Wing Design Application

Rom the theoretical point of view it is rather standard routine paper, where the finite element (FE) model is developed to study the structural performance. The value and the main advantage of this paper is high complexity of simulation object – the wing on an unmanned flying vehicle. Numerical analysis is supported by experimental. The paper is  well written and contains interesting details

Author Response

Thank you for your comments. And we acknowledged it.

Reviewer 3 Report

The authors present a good work on composite modelling. The modelling process is initially validated and the work carried out is technically sound. Please see my queries in the attached report. If the authors perform the required suggestions, this work can add value to the scientific community.

Round 2

Reviewer 1 Report

Still some unanswered question and flaws in the manuscript. Are the authors familiar with the yield criterion? If so, then please use expression "von Mises stress", with the correct name.
